# Soil community composition in dynamic stages of semi-natural calcareous grassland

**A. Y. Ayesh Piyara Wipulasena**[ID]*, **John Davison, Aveliina Helm**[ID], **Liis Kasari, Mari Moora, Elisabeth Prangel, Triin Reitalu, Tanel Vahter, Martti Vasar, Martin Zobel**

Department of Botany, Institute of Ecology and Earth Sciences, Faculty of Science and Technology, University of Tartu, Tartu, Estonia

* ayesh.wipulasena@ut.ee

**Data Availability Statement:** The data used in this study have been deposited in the NCBI SRA and can be accessed using the accession number BioProject PRJNA957953.

## Abstract

European dry thin-soil calcareous grasslands (alvars) are species-rich semi-natural habitats. Cessation of traditional management, such as mowing and grazing, leads to shrub and tree encroachment and the local extinction of characteristic alvar species. While soil microbes are known to play a critical role in driving vegetation and ecosystem dynamics, more information is needed about their composition and function in grasslands of different dynamic stages. Here we assess the composition of soil fungal, prokaryotic, and plant communities using soil environmental DNA from restored alvar grasslands in Estonia. The study areas included grasslands that had experienced different degrees of woody encroachment prior to restoration (woody plant removal and grazing), as well as unmanaged open grasslands. We found that, in general, different taxonomic groups exhibited correlated patterns of between-community variation. Previous forest sites, which had prior to restoration experienced a high degree of woody encroachment by ectomycorrhizal Scots pine, were compositionally most distinct from managed open grasslands, which had little woody vegetation even prior to restoration. The functional structure of plant and fungal communities varied in ways that were consistent with the representation of mycorrhizal types in the ecosystems prior to restoration. Compositional differences between managed and unmanaged open grasslands reflecting the implementation of grazing without further management interventions were clearer among fungal, and to an extent prokaryotic, communities than among plant communities. While previous studies have shown that during woody encroachment of alvar grassland, plant communities change first and fungal communities follow, our DNA-based results suggest that microbial communities reacted faster than plant communities during the restoration of grazing management in alvar grassland. We conclude that while the plant community responds faster to cessation of management, the fungal community responds faster to restoration of management. This may indicate hysteresis, where the eventual pathway back to the original state (grazed ecosystem) differs from the pathway taken towards the alternative state (abandoned semi-natural grassland ecosystem).

**Funding:** This study was funded by the Estonian Research Council (https://etag.ee/en/) (PRG1065 to MM, MZ, JD, MV and AYAPW, PRG1789 to TV, PRG874 to AH, TR, LK, and EP) and European Commission LIFE+ pro-gramme (https://cinea.ec. europa.eu/programmes/life_en) (projects LIFE to Alvars LIFE13 NAT/EE/000082 and ForEst&FarmLand LIFE18IPE/EE/000007 to AH, TR, LK, and EP). The funders had no role in study de-sign, data collection and analysis, decision to publish, or preparation of the manuscript.

**Competing interests:** The authors have declared that no competing interests exist.

## Introduction

There is growing interest in applying alternative stable state theory to explain major ecosystem patterns [1, 2]. For instance, arid ecosystems can have different stable states, including savannah and forest, depending on the co-effects of precipitation, fire, and herbivory [2]. In temperate semi-natural ecosystems, the state of the ecosystem—grassland or forest—depends on the management regime [3]. However, there are also ecosystems, notably scrub of different densities, that represents the transition from stable forest to stable grassland or *vice versa* [4, 5]. In such a case, scrub ecosystems may themselves represent alternative transitional trajectories either from forest to grassland (generation of semi-natural grassland due to cutting woody plants, making hay, and grazing with domestic animals) or from grassland to forest (woody plant encroachment in abandoned grasslands). Transitional shrubby ecosystems vary in structure (species composition and diversity) and/or function (total biomass and carbon flux) due to differences in immigration history, disturbance history, or other stochastic processes [6].

Although nature-based climate strategies worldwide have focused on tree-planting, there is increasing awareness of the multiple ecosystem services provided by biodiverse grasslands [7, 8]. Significant land use changes since the middle of the last century have resulted in many productive European semi-natural grasslands being converted to cultivated grassland and arable fields [9] and other less productive grasslands being abandoned [10]. Cessation of management results in succession towards forest, causing shrub and tree encroachment and the loss of typical grassland species [11, 12].

Alvars are highly diverse grasslands occurring on thin soil (generally <20 cm) over calcareous bedrock in the Baltic Sea region [13, 14]. Though alvar plant species have been present in Europe during much of the post-glacial period [15, 16], most current communities are semi-natural. This means that they are maintained by human activities such as mowing, grazing and cutting woody vegetation, mainly Scots pine (*Pinus sylvestris*) [17]. The area of Estonian alvar grasslands has declined during the last century, either due to abandonment and subsequent woody plant encroachment or conversion to agricultural land [12]. However, in the last decade, restoration projects have aimed to remove woody encroachment from Estonian alvars and re-commence grazing with domestic animals. Recovering species composition in restored grassland communities usually relies on natural recruitment, either from the seed rain or seed bank [18].

In recent years, the significance of soil microbes in driving plant community composition has been recognised [19]. Mutualistic and antagonistic microbes considerably alter plant performance [20, 21], and some effects of environmental change on plant communities are mediated by plant-microbe interactions [21, 22]. The importance of microbes is also acknowledged in restoration ecology [23–26], where the absence of specific microbial groups has been observed to retard the succession of entire communities and introduction of mutualistic microbes may facilitate restoration [27–30].

The degree to which aboveground plant community dynamics mirror changes in soil microbial communities is not clearly understood. In general, the diversities of different taxonomic and functional groups tend to be positively correlated [31]. Sepp et al. (2021) [32] addressed the alternative stable states of forested and open patches in a calcareous wooded meadow and found generally good coincidence between plant and microbial community compositional patterns. Therefore, it might be assumed that aboveground vegetation and soil microbial communities co-vary in response to changes in management regimes or in natural conditions such as climate. However, it is also possible that the rates of change vary between taxonomic groups and that changes in one group regularly precede or follow changes in another group. It has been hypothesised that earlier changes in certain groups may indicate

that these groups actively drive changes in other groups [33, 34]. For instance, where community turnover among plants precedes that of their symbionts, arbuscular mycorrhizal (AM) fungi [35], we may hypothesise that plants drive changes in mycorrhizal fungal communities. To further address these hypotheses, an experimental approach is needed.

As indicated above, alternative transient trajectories may emerge during ecosystem change between stable states such as natural forest and traditionally managed semi-natural grassland. The predominant mycorrhizal type of the ecosystem has the potential to be a characteristic that influences the trajectory between stable states. First, ectomycorrhizal (ECM) fungi represent an exception to the general pattern that diversity of different taxonomic groups varies in parallel—ECM diversity correlates negatively with that of plants and arbuscular mycorrhizal (AM) fungi at local and global scales [32, 36]. Second, while alvar vegetation consists mainly of AM plant species, cessation of management causes a progressive increase in the dominance of ECM trees. Because ECM trees strongly influence the rest of the plant community [19] and often suppress plant [36] and AM fungal diversity [19], it is conceivable that the extent of ECM dominance influences transitional trajectories among dynamic plants and soil microbial communities.

Here, we compared four dynamic stages of alvar grassland in western Estonia, representing restored grasslands that had experienced different successional histories prior to the start of restoration management. Restoration management took the form of woody vegetation removal and re-establishment of grazing with domestic animals. The four dynamic stages were as follows: 1) open treeless semi-natural grassland vegetation with some junipers and occasional pine individuals where restoration management was not implemented, 2) open treeless semi-natural grassland vegetation where restoration management was implemented, 3) juniper scrub encroachment with single pine individuals on formerly open semi-natural grassland where restoration management was implemented, and 4) young pine forest on formerly open semi-natural grassland where restoration management was implemented. We extracted environmental DNA (eDNA) from the soil and used metabarcoding to characterise fungal (18S rRNA gene), prokaryotic (16S rRNA gene) and plant (P6 loop of the plastid trnL; UAA) communities. First, we hypothesised that plant, soil fungal and soil prokaryotic communities vary between stages in approximately the same way, such that compositional turnover between dynamic stages is similar irrespective of which taxonomic group is addressed. Second, we hypothesised that the history of ECM tree dominance influences the trajectory of the dynamics of plant and soil microbial communities in response to changed management regime. Thus, grasslands with a recent history of ECM dominance should differ most clearly from those with a recent history of predominantly AM vegetation. Third, we hypothesised that taxonomic variation between restored grasslands mirrors the variation in functional group composition.

## Methods

### Study area

Sampling was conducted in 2019 in alvar habitats in western Estonia, including the mainland and three larger islands in the Baltic Sea—Saaremaa, Hiiumaa, and Muhu (Fig 1, S1 Table). Alvars are semi-natural grasslands on Silurian or Ordovician calcareous bedrock characterised by shallow soils (up to 20–30 cm) [13]. We used stratified random selection of study areas, i.e. selecting 29 sites randomly within one vegetation type, to assure that they were as similar as possible to each other in terms of vegetation and ecological conditions. We addressed grasslands of the type 'Avenetum alvarense' described by Pärtel et al. (1999) [13], which is the most widespread alvar vegetation type. The soil is strongly calcareous, with pH>7, and a humus layer of <20 cm overlays the weathered limestone bedrock. Characteristic herbaceous plant

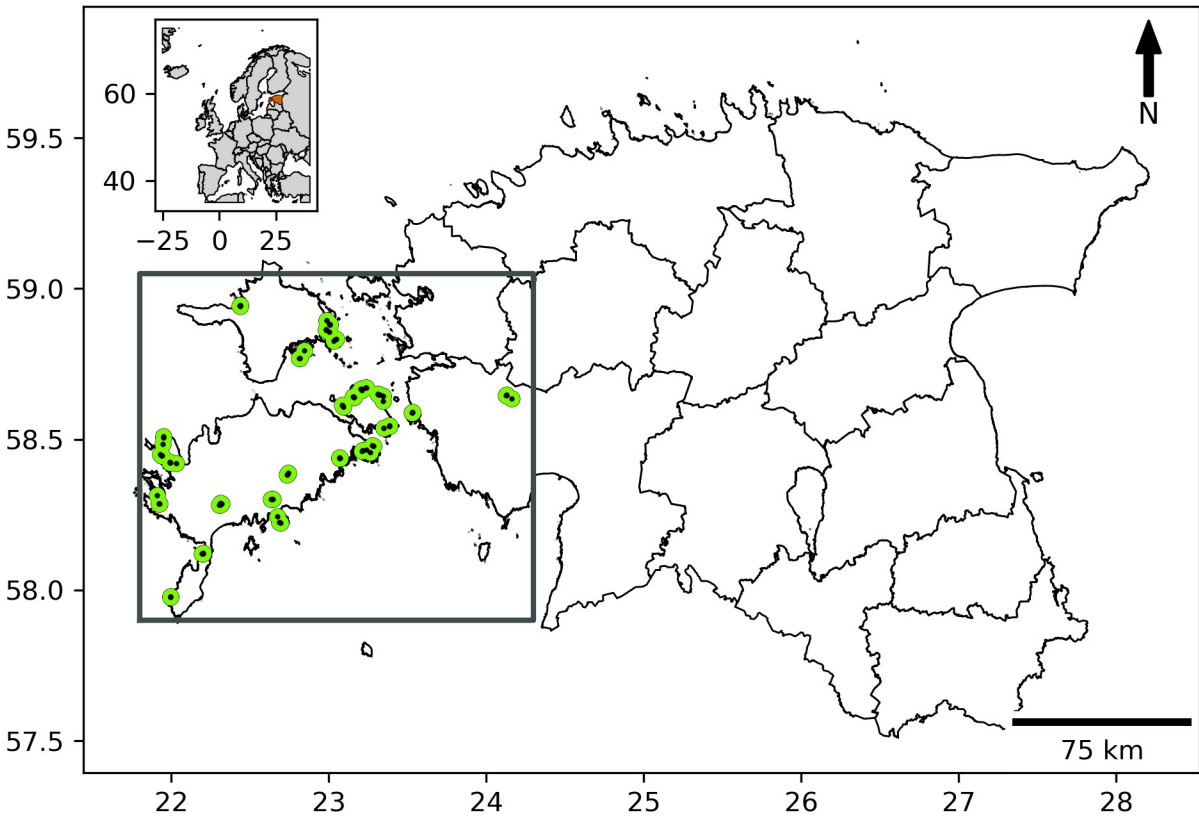

**Fig 1. Map of study sites in western Estonia.** Each site comprised four dynamic stages reflecting the degree of woody encroachment experienced prior to restoration: unmanaged open, managed open, managed previous scrub, and managed previous forest (see methods for details). The map was made using freely available vector and raster map data from Natural Earth (naturalearthdata.com) and the Estonian Administrative and Settlement Division: Land Board; accessed 01.06.2023.

species include—*Helictotrichon pratense*, *Carex tomentosa*, *Briza media*, *Festuca ovina*, *Sesleria caerulea*, and *Centaurea jacea* (see also Prangel et al. 2023 [12] for details about the environmental conditions at plots). We obtained permission for the study from the Estonian Environmental Board (11.02.2015 nr 17–2.1/15/2415-2).

At each of the 29 study sites, we selected four subsites representing different dynamic stages of grassland ecosystems on dry calcareous soils, avoiding temporarily wet areas and areas with limestone outcrops. Specifically, we selected (i) unmanaged open: continuously open grassland with no management change (no mowing or grazing); (ii) managed open: continuously open grassland, with grazing reinstated in 2015; (iii) managed previous scrub: scrub removed and grazing reinstated in 2015; (iv) managed previous forest: forest removed and grazing reinstated in 2015. All areas where grazing grassland management was reinstated were restored as a part of the European Commission LIFE+ Programme project LIFE to Alvars [37]. At each subsite, a total of 116 plots, a 5 g topsoil sample was collected from a randomly-selected small 10x10 cm quadrat near the corner of each plot. After removing plant roots from the soil, samples were dried for 24 h at 50˚C and stored dry prior to molecular analysis.

## Molecular analysis

DNA was extracted from 5 g of dried topsoil using the PowerMax Soil DNA Isolation Kit. The 16S rRNA gene was used to identify prokaryotes, the 18S rRNA gene was used to identify

eukaryotes (with the ultimate goal of identifying fungi), and the short fragment of the trnL (UAA) intron (the P6 loop, 10–143 bp) was used to identify plants. The DNA metabarcoding approach follows the methods described by Vasar et al. 2022 [38]. Briefly, for prokaryotes, primers 515F and 926R were used to amplify the 16S rRNA variable V4 region [39, 40]; for eukaryotes, primers Euk575F and Euk895R were used to amplify the 18S rRNA region [41, 42]; and for plants primers trnL-g and trnL-h were used to amplify the short fragment of the trnL (UAA) intron (the P6 loop) [43]. The 16S rRNA, 18S rRNA and trnL amplicons were sequenced on the Illumina MiSeq platform (using a 2 × 250 bp, 2 × 250 bp and 2 x 150 bp paired-read sequencing approaches, respectively; Asper Biogene [Tartu, Estonia]).

## Bioinformatics

Bioinformatic processing of sequencing data was conducted using the gDAT pipeline [44]. There were 2x9 973 435 raw prokaryotic 16S reads, from which 5 738 429 reads could be cleaned and combined. Reads were clustered at 97% similarity, yielding 107 335 OTUs, of which 96 144 were chimera free. Obtained OTUs were subjected to BLAST search [45] against the SILVA 16S database (v138.1, Pruesse et al., 2007) [46]. For robust assignment of taxonomic identity, the lowest common ancestor (LCA) approach with multiple best-hits was used to build a consensus taxonomy where at least 51% of hits should contain the same identity at the accepted taxonomic rank. FAPROTAX [47] was used to assign taxa to functional guilds, with the guild classification simplified following Vasar et al., (2022) [38].

There were 2x3 761 473 raw eukaryotic 18S reads, from which 3 176 172 reads could be cleaned and combined. Reads were clustered with 97% similarity, yielding 32 388 OTUs, of which 29 105 were chimera free. Obtained OTUs were subjected to BLAST search against the NCBI non-redundant nucleotide database [48] using LCA with the same criteria as for 16S reads. Fungal OTUs were distinguished and divided into functional groups using the Fungal-Traits database [49].

There were 2x18 765 961 raw trnL reads, from which 13 052 012 reads could be cleaned and combined. Reads were clustered at 97% similarity, yielding 54 232 OTUs, of which 48 809 were chimera free. Obtained OTUs were subjected to BLAST search against the NCBI non-redundant nucleotide database using LCA with the same criteria as for 16S reads. Plants were classified into forbs, graminoid, woody and legumes.

For all datasets, sequences were assigned to taxonomic levels with BLAST using 80% alignment thresholds and the following criteria for different taxonomic ranks: families at >90%, genera at >95%, and species at >97% sequence similarity. Raw reads from this Targeted Locus Study have been deposited in the NCBI SRA (BioProject PRJNA957953).

## Statistical analysis

**Data cleaning.** Each data set was cleaned to remove samples with < 1000 reads and singleton OTUs. This left 101 samples for prokaryotes, 58 samples for fungi and 106 samples for plants (S1 Table). The cleaned data matrices were normalised using variance stabilising transformation (VST) (using the R package DESeq2 version 1.38.3: Love et al. 2014) [50], as suggested by McMurdie and Holmes (2014) [51]. The method uses fitted dispersion–mean relationships to normalise data with respect to sample size (sequencing depth of individual samples) and variance.

**Alpha diversity.** The alpha diversity of soil communities was assessed by calculating richness in VST-transformed data. Differences in richness between different dynamic stages were assessed using linear mixed-effects models, with a random intercept associated with the site (R package lme4 version 1.1–32). Asymptotic richness (Chao index), asymptotic Shannon

diversity index and asymptotic Simpson diversity were also calculated from cleaned raw sequencing data (i.e. prior to VST transformation) using the iNEXT R package (version 3.0.0; Chao et al. 2016) [52] and modelled in the same manner.

**Beta diversity.** The effects of dynamic stages on the community structure (beta-diversity) of plants and different soil organisms were assessed with distance-based redundancy analysis (dbRDA) using Bray-Curtis distance and VST-transformed data (dbRDA from R package vegan; Oksanen et al. 2022) [53]. Hypotheses were tested using within-site permutations (999 permutations). Non-Metric Multidimensional Scaling (NMDS) of the VST-transformed data was also used for visualisation. Pairwise correlations between different communities of organisms were estimated using Procrustes rotation. The rotation was carried out using the principal coordinates of Bray-Curtis distance matrices (VST-transformed data) containing common samples (with the number of retained axes equal to the minimum number of positive eigenvalues derived from the matrices being compared). Differences between dynamic stages in the magnitude of Procrustes residuals were tested using linear mixed models with the structure described above.

**Taxonomic and functional groups.** The relative abundances of different phyla and functional guilds of organisms were estimated from transformed read counts and OTU counts in the VST-transformed data. For prokaryotes, the abundance of organisms assigned to multiple functional groups was duplicated for each category. dbRDA was used to assess differences in the functional and taxonomic composition of different dynamic stages using the restricted permutation procedure described above. Differences between habitats in the relative abundance and richness of individual guilds were estimated using linear mixed models with the structure described above. dbRDA was also used to assess the effect of the dynamic stage on the community structure of individual guilds.

## Results

The alpha diversity of prokaryotes and fungi did not differ significantly between dynamic stages (Fig 2). The alpha diversity of plants (as measured from eDNA samples) was higher in the unmanaged open habitat than in all other stages (Fig 2). Very similar results were produced by the asymptotic alpha diversity estimates, though significantly lower asymptotic Shannon diversity was recorded in managed previous forest for prokaryotes (S1 Fig).

Dynamic stage had a significant effect on the community composition of all organism groups (Fig 3, S2 Fig). In all cases, the managed open and managed previous forest stages were most distinct, with managed previous scrub and unmanaged open stages intermediate (Fig 3).

Procrustes rotation indicated a strong correlation in the compositional variation of different organism groups: strongest correlation was observed between prokaryotes and fungi, while relationships involving plants were weaker (Table 1). No significant differences between dynamic stages were observed in the magnitude of Procrustes residuals (Table 1).

No clear differences between dynamic stages were observed in the functional composition of prokaryotes (dbRDA $R^2$ = 0.021, P = 0.575; Fig 4). The abundance of different fungal guilds differed between habitats (dbRDA $R^2$ = 0.166, P = 0.001; Fig 4), with AM fungi more abundant in managed open grassland habitat and ECM fungi more abundant in the managed previous forest and unmanaged open sites. There were also differences between stages in the abundance of plant groups (dbRDA $R^2$ = 0.079, P = 0.014; Fig 4) with more woody plants in the managed previous forested and unmanaged open sites, and more forbs and legumes in the managed open site. The broad taxonomic composition of the different communities in different dynamic stages is shown in S3 Fig.

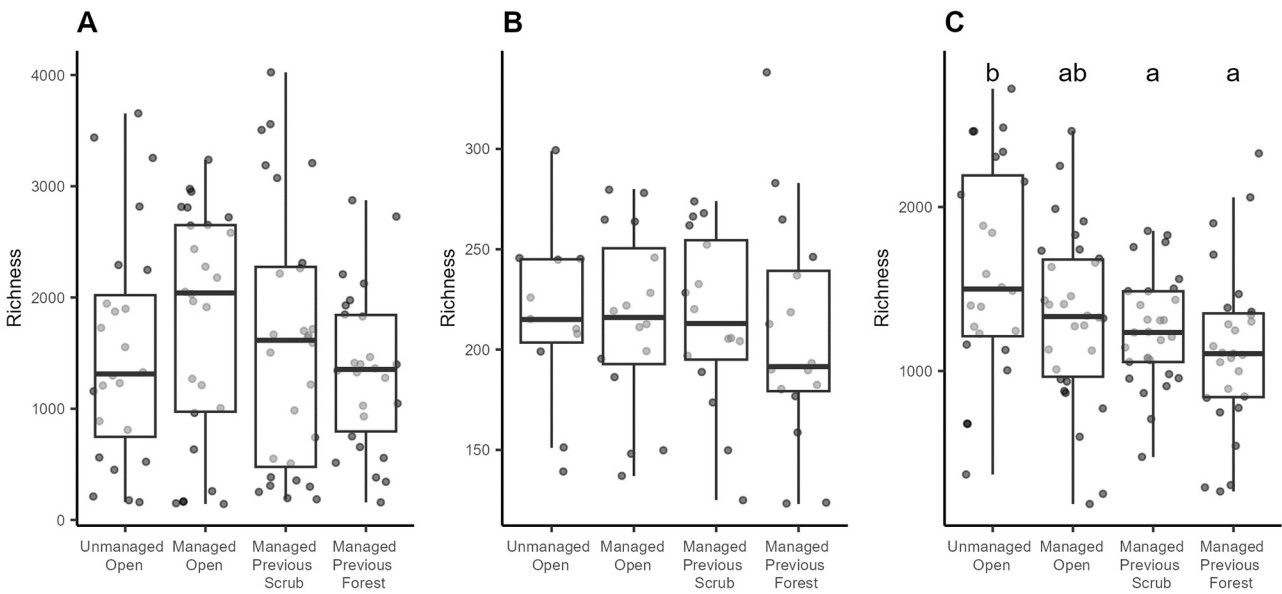

**Fig 2. The richness of prokaryotic (A), fungal (B) and plant (C) communities (VST-transformed data matrices).** Letters above boxes indicate significant differences between groups.

There were no differences between dynamic stages in the richness of any prokaryotic guild (S4 Fig). Among fungi, AM fungi were significantly richer in the managed open than in the managed previous forest stage (S5 Fig). Woody plants were significantly less rich in the managed open stage compared with all other stages (S6 Fig). Richness of forbs was lowest in the managed previous forest stage, highest in the unmanaged open stage, and intermediate in the other stages (S6 Fig). Legume richness was lower in managed previous forest than in all other stages (S6 Fig). Within-guild composition varied between stages for all guilds (S7–S9 Figs). In general, compositional differences were analogous to those seen for entire groups (Fig 3). However, in the managed and unmanaged open habitats, AM fungal, fungal parasite and graminoid communities were notably similar (S8 and S9 Figs).

## Discussion

In recent years, eDNA has been used in the description of communities, as it allows the parallel study of different taxonomic groups [38]. We recorded strong correlations in the composition of plant, soil fungal, and soil prokaryotic communities among transient dynamic stages of alvar grassland in habitats that had reached varying degrees of woody encroachment prior to restoration management. Nonetheless, between-stage differences in fungal community composition were larger than the corresponding differences in plant and prokaryotic communities; and while the alpha diversity of plants varied between stages, those of fungi and prokaryotes did not. Community composition in managed previous forest was most distinct from that in managed open grassland for all organism groups, with managed previous scrub and unmanaged grassland intermediate.

The synchronised pattern of changes in plant, fungal and prokaryote communities generally supports our first hypothesis and suggests that different groups of interacting organisms all track changes in management regime and accompanied ecological conditions, as predicted by the habitat hypothesis by Zobel & Öpik (2014) [34]. To a certain extent, the similarity

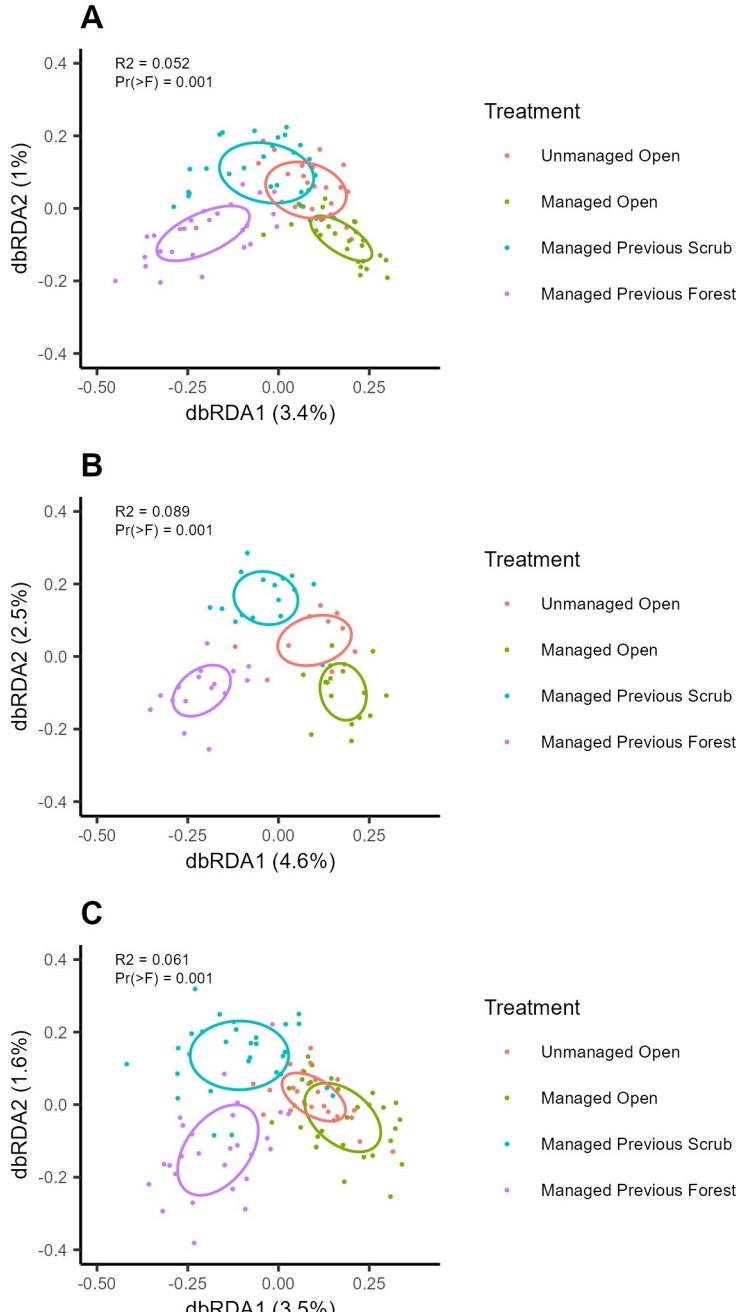

**Fig 3. Distance-based redundancy analysis of prokaryotic (A), fungal (B) and plant (C) community composition.**

between different dynamic stages might reflect the fact that DNA persists in the soil for several years and eDNA based community descriptions incorporate both present and past community composition [32, 54]. However, a direct comparison of managed and unmanaged open grassland areas revealed a slightly different picture. Specifically, the unmanaged open areas represent the starting state of the managed open area before restoration management was initiated. Although the alpha diversity of managed and unmanaged open ecosystems did not differ, there were differences in community composition, and the different taxonomic groups varied

**Table 1. Procrustes correlation between the composition of prokaryotic, fungal and plant communities, and analysis of Procrustes residuals.**

| | | Protest | | | ANOVA on residuals | | | |
|---|---|---|---|---|---|---|---|---|
| Group | Intersect samples | Procrustes Sum of Squares (m12 squared) | Correlation in a symmetric Procrustes rotation | Significance | F | Df | Df.res | Pr (>F) |
| Prokaryotes and Fungi | between each other (k = 53) | 0.1223 | 0.9368 | 0.001 | 0.522 | 3 | 39.906 | 0.670 |
| Fungi and Plants | between each other (k = 58) | 0.2330 | 0.8758 | 0.001 | 1.651 | 3 | 46.158 | 0.191 |
| Prokaryotes and Plants | between each other (k = 103) | 0.3408 | 0.8119 | 0.001 | 2.181 | 3 | 77.066 | 0.097 |

in the degree of divergence, with relatively clearer differences in fungal, and to a degree, prokaryotic composition, compared with plant composition. This might suggest that microbial communities respond more quickly than plants to active restoration management. When changes in certain microbial communities, such as arbuscular mycorrhizal fungi, precede those of plants, it may indicate that the quicker responders, i.e. microbes, act as 'drivers' while plants are 'passengers' [33]. However, a more comprehensive understanding of which is driver and which is passenger in the case of a change in the biotic community can only be obtained experimentally.

The pattern indicated by the results of this study—that fungi respond quicker than plants—is the opposite of Neuenkamp et al. (2018)'s [35] finding that plants preceded AM fungi during woody plant encroachment succession in alvar grassland. In principle, the difference might reflect hysteresis. Thus, changes in one direction towards a certain ecosystem state differ substantially from changes in the reverse direction. Work in other ecosystems has also shown that the movement of the ecosystem away from a certain state proceeds differently than the movement back to the original state [55, 56]. However, this had not previously been investigated for systems including microbes. In alvar ecosystems, it may thus be hypothesised that during woody plant encroachment, plants drive the changes, while in the case of management-induced changes in the opposite direction, microbes drive the changes. On the other hand, the slower response of plants might simply reflect more efficient dispersal of some microbes compared with plants [57, 58] or a better ability of some plants to persist belowground during unsuitable periods [59]. The current data are insufficient to distinguish these possibilities and draw general conclusions, but they reveal relationships that deserve attention in further studies.

We also hypothesised that the history of ECM trees in the ecosystem influences current plant and soil microbial communities. This legacy was observed in all measured community types. The communities in managed previous forested (with ECM pines) and managed open grasslands were always most distinct from each other. Communities in previous scrub and unmanaged open grassland, where a historical or contemporary absence of management allowed some pine individuals to establish, were intermediate between the previously forested and managed open stages. ECM vegetation can have a profound influence on multiple biotic and abiotic ecosystem properties. For instance, ECM may diminish the negative effect of competition with AM plants on ECM plants when ECM fungal mycelia lock up a substantial proportion of soil nutrients, particularly starving saprotrophic fungi and co-occurring AM plants of nutrients [60]. ECM fungi also form efficient mycorrhizal networks to enhance seedling establishment of ECM plants [61] and form a physical sheath around the young feeder roots of trees, offering protection from herbivores and pathogens [62, 63]. Plant communities with ECM trees exhibit lower species richness than predominantly AM plants-dominated

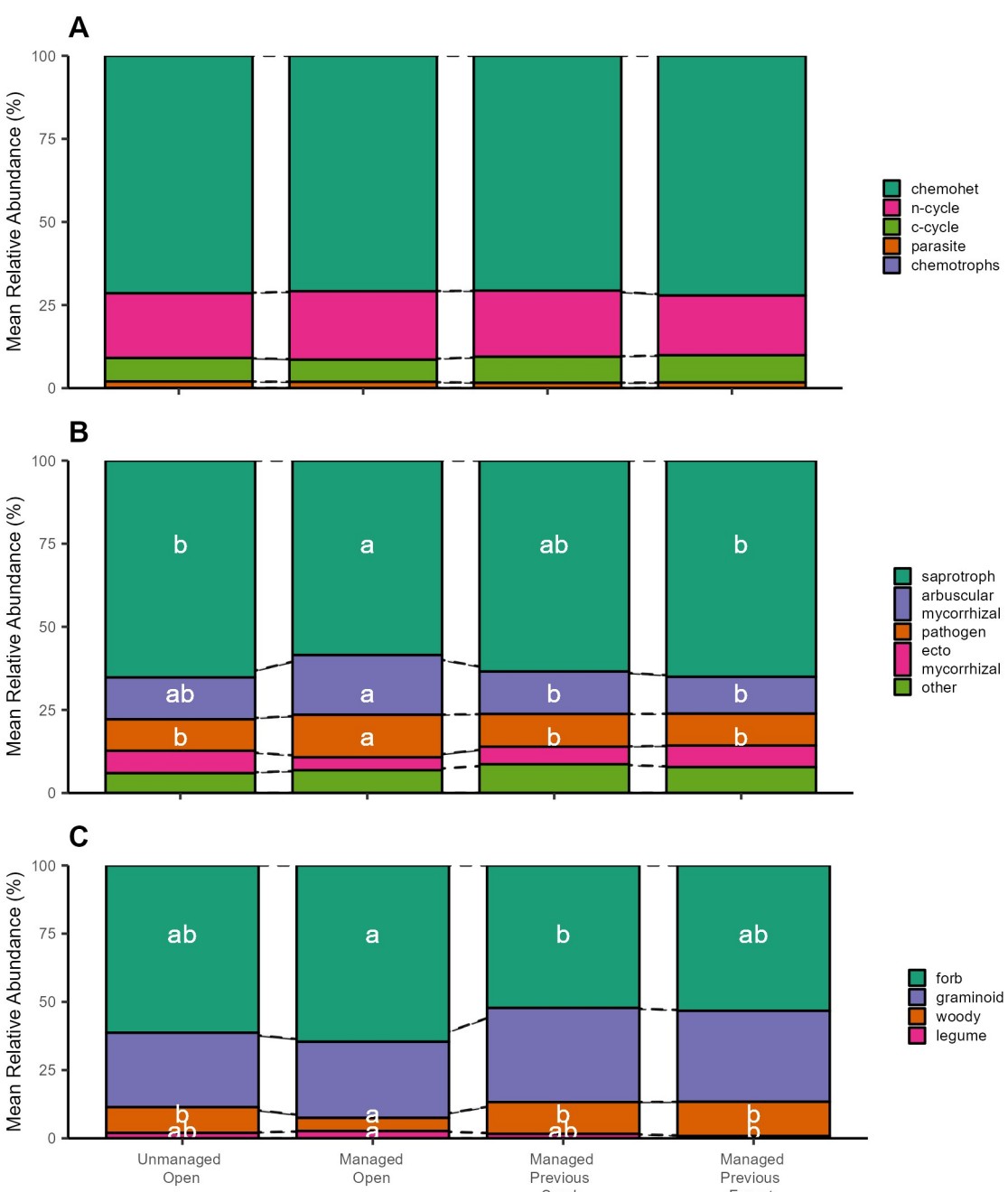

**Fig 4. Mean relative abundances (%) of the different functional guilds of prokaryotes (A), fungi (B) and plants (C) in different dynamic stages.** Letters within boxes indicate significant differences between groups within individual functional groups.

vegetation [36]. Thus, woody plant dominance generally has an effect on the entire biota, and some of this effect is most likely specific to ECM trees.

In general support of our third hypothesis, we found that compositional changes among plants and fungi in different dynamic stages reflected shifts in functional composition, while the function of prokaryotic communities did not change measurably. It is unclear whether the latter result reflects genuine functional homogeneity or, rather, the resolution of the

prokaryotic functional classification [47]. However, the functional changes in plant and fungal communities were logical and consistent with the hypothesised dynamics of different mycorrhizal types. Notably, woody plants were markedly less abundant in the managed open grasslands, and, albeit variably, ECM fungi showed the same overall pattern. Conversely, AM fungi were most abundant in the managed open stage. This variation is consistent with the notion that transitions towards a stable habitat state may take different functional trajectories depending on the starting point.

At the coarse scale, we conclude that different taxonomic groups exhibited similar patterns of variation between dynamic habitat stages. However, at the finer scale, subtle differences between groups and dynamic stages were apparent. One of the differentiators was the occurrence of woody vegetation in general and of ECM plants and fungi in particular. Analogous differences were found in a previous study by Neuenkamp et al. (2018) [35] who showed that in the case of woody encroachment into the alvar grassland ecosystem, plant communities change with fungal communities responding later. In this study, however, we found that fungal communities seemed to react more rapidly, while the response of the plant communities was slower. We suggest that unequal speed of change among different functional groups might indicate that the rapidly changing group is driving change in the slower group. Moreover, this kind of dynamics, where the pathway from one stable state, managed alvar grassland, to another, alvar forest, differs from the pathway taken in the opposite direction (from the alvar forest towards the 'original' stable state of alvar grassland) may indicate hysteresis [56]. However, experimental study is needed to definitively understand these relationships.

## Supporting information

**S1 Table. Experimental sites (29 sites)—After cleaning.**
(PDF)

**S1 Fig. The asymptotic richness, Shannon diversity and Simpson diversity of the raw community matrices of the prokaryotes (A), fungi (B) and plants (C).** The letters above the boxes indicate significant differences between groups.
(TIF)

**S2 Fig. Non-metric multidimensional scaling of prokaryotes (A), fungi (B) and plants (C) community data (VST-transformed).**
(TIF)

**S3 Fig. The mean relative abundances (%) of the major taxonomic groups (phyla for fungi (B) and prokaryotes (A); orders for plants (C)) in different habitats.**
(TIF)

**S4 Fig. The richness of different prokaryotic guilds in different habitats.**
(TIF)

**S5 Fig. The richness of different fungal functional guilds in different habitats.** The letters above the boxes indicate significant differences between groups.
(TIF)

**S6 Fig. The richness of different plant groups in different habitats.** The letters above the boxes indicate significant differences between groups.
(TIF)

**S7 Fig. Distance-based redundancy analysis of prokaryotic functional guilds.**
(TIF)

**S8 Fig. Distance-based redundancy analysis of fungal functional guilds.**
(TIF)

**S9 Fig. Distance-based redundancy analysis of plant functional guilds.**
(TIF)

## Acknowledgments

The authors thank Lena Neuenkamp and the University of Tartu Landscape Biodiversity workgroup members and students for their help during the fieldwork.

## Author Contributions

**Conceptualization:** John Davison, Aveliina Helm, Mari Moora, Martin Zobel.

**Data curation:** Liis Kasari, Elisabeth Prangel, Triin Reitalu.

**Formal analysis:** A. Y. Ayesh Piyara Wipulasena, John Davison, Martti Vasar.

**Funding acquisition:** Aveliina Helm, Mari Moora.

**Methodology:** Aveliina Helm, Liis Kasari, Elisabeth Prangel, Triin Reitalu.

**Project administration:** Aveliina Helm.

**Supervision:** John Davison, Mari Moora, Tanel Vahter, Martin Zobel.

**Visualization:** A. Y. Ayesh Piyara Wipulasena.

**Writing – original draft:** A. Y. Ayesh Piyara Wipulasena, John Davison, Martin Zobel.

**Writing – review & editing:** A. Y. Ayesh Piyara Wipulasena, John Davison, Aveliina Helm, Liis Kasari, Mari Moora, Elisabeth Prangel, Triin Reitalu, Tanel Vahter, Martti Vasar, Martin Zobel.

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
