## [Decision Letter · Decision Letter 0]

2 Aug 2023

PONE-D-23-21042Soil community composition in dynamic stages of semi-natural calcareous grasslandPLOS ONE

Dear Dr. Wipulasena,

Thank you for submitting your manuscript to PLOS ONE. After careful consideration, we feel that it has merit but does not fully meet PLOS ONE’s publication criteria as it currently stands. Therefore, we invite you to submit a revised version of the manuscript that addresses the points raised during the review process.

We look forward to receiving your revised manuscript.

Kind regards,

Jian Liu

Academic Editor

PLOS ONE

Journal Requirements:

   "The authors thank Lena Neuenkamp and the University of Tartu Landscape Biodiversity workgroup members and students for their help during the fieldwork. This study was funded by the Estonian Research Council (PRG1065 and PRG874 and European Commission LIFE+ programme (projects LIFE to Alvars LIFE13 NAT/EE/000082 and ForEst&FarmLand LIFE18IPE/EE/000007)."

   "This study was funded by the Estonian Research Council (https://etag.ee/en/) (PRG1065 to MM and PRG874 to AH and European Commission LIFE+ programme (https://cinea.ec.europa.eu/programmes/life_en) (projects LIFE to Alvars LIFE13 NAT/EE/000082 to AH and ForEst&FarmLand LIFE18IPE/EE/000007 to AH).

5. We note that Figure 1 in your submission contain map/satellite images which may be copyrighted. All PLOS content is published under the Creative Commons Attribution License (CC BY 4.0), which means that the manuscript, images, and Supporting Information files will be freely available online, and any third party is permitted to access, download, copy, distribute, and use these materials in any way, even commercially, with proper attribution. For these reasons, we cannot publish previously copyrighted maps or satellite images created using proprietary data, such as Google software (Google Maps, Street View, and Earth). For more information, see our copyright guidelines: http://journals.plos.org/plosone/s/licenses-and-copyright.

Additional Editor Comments:

The study is interesting and the manuscript is relatively well written. But the manuscript still has some problems as suggested by the reviewers. The authors should respond to the comments of the reviewers one by one and revise the manuscript accordingly.  

Reviewers' comments:

Reviewer's Responses to Questions

**Comments to the Author**

1. Is the manuscript technically sound, and do the data support the conclusions?

Reviewer #1: Yes

Reviewer #2: Yes

2. Has the statistical analysis been performed appropriately and rigorously? 

Reviewer #1: Yes

Reviewer #2: Yes

3. Have the authors made all data underlying the findings in their manuscript fully available?

Reviewer #1: Yes

Reviewer #2: Yes

4. Is the manuscript presented in an intelligible fashion and written in standard English?

Reviewer #1: Yes

Reviewer #2: Yes

5. Review Comments to the Author

Reviewer #1: Reviewer comments to the ms entitled “Soil community composition in dynamic stages of semi-natural calcareous grassland”

Overall the manuscript is well-written and concise. It covers and important topic and should be published after some minor issues are treated and addressed.

l56 – “stable states” – what do you mean with this?

l126-133 The hypotheses are a bit foggy and based on my opinion not very well introduced in the introduction section. The community types studied are not clearly introduced here – it is not clearly said that the effect of restoration works on variously long-lasting encroachment and reference unmanaged grasslands are compared here. In the first hypotheses is not clear what do you mean with “communities vary in parallel”

l338-340 This assumption is rather strong. If something changes faster than the other it does not really means that it driver anything – it could be just simply that this type of community/organism group is more sensitive to the changes than the other. Of course in case of shrub encroachment because of direct competition for light you can imply that the shrub encroachment drives the changes in the herb layer, but this is also highly resistant to a particular level of encroachment. This assumption should be down toned a bit.

Reviewer #2: This paper compared variations of soil microbial community composition and vegetation composition following different stages of woody encroachment and analyzed the possible drivers of these variations. The study may provide valuable information for management of semi-natural calcareous grassland.

Generally, the manuscript was well written. The introduction was logically organized and the discussion part provided sufficient analysis and interpretation of the data. The following are two suggestions for the authors.

（1） The definite conclusions seemed to be missing in both the abstract and in the main text.

（2） In the Methods part, there were no sufficient information on the vegetation (species identity, abundance, canopy height, etc.) , climate, and soil etc. for each site sampled.

6. PLOS authors have the option to publish the peer review history of their article (what does this mean?). If published, this will include your full peer review and any attached files.

Reviewer #1: No

Reviewer #2: No

---

## [Author Response · Author response to Decision Letter 0]

10 Aug 2023

PONE-D-23-21042

Soil community composition in dynamic stages of semi-natural calcareous grassland

Dr Jian Liu

Academic Editor

PLOS ONE

Dear Editor,

Thank you very much for sending reviewer comments and an editorial decision on our manuscript. We are extremely grateful to you and the reviewers for the very useful comments. We have considered all of the comments and have now prepared a revised version that we believe is improved and manages to overcome the issues noted by the reviewers.

Please find below a full description of our responses (in regular typeface) to every editor or reviewer comment (in italics).

The academic editor

In your Methods section, please provide additional information regarding the permits you obtained for the work. Please ensure you have included the full name of the authority that approved the field site access and, if no permits were required, a brief statement explaining why.

We have added a reference to the permission' Estonian Environmental Board 11.02.2015 nr 17-2.1/15/2415-2' in line number 185 of the manuscript with track changes.

We note that Figure 1 in your submission contains map/satellite images which may be copyrighted…. We require you to either (a) present written permission from the copyright holder to publish these figures specifically under the CC BY 4.0 license or (b) remove the figures from your submission.

The map was generated using open-source GIS data. The main plot has data from the Estonian Coun-ties sourced from the Estonian Land Board (https://geoportaal.maaamet.ee/est/Ruumiandmed/Haldus-ja-asustusjaotus-p119.html), and the inner map has data sourced from Natural Earth (https://www.naturalearthdata.com/about/terms-of-use/). No proprietary or copyrighted data or applica-tions were used to generate the plot. We have added information about the data sources to the legend of Figure 1.

Please review your reference list to ensure that it is complete and correct.

Thank you, done.

Reviewer #1: 

Overall the manuscript is well-written and concise. It covers and important topic and should be published after some minor issues are treated and addressed.

Thank you!

l56 – "stable states" – what do you mean with this?

We mean 'the successional stability of ecosystems' and included a more thorough explanation of ecosystem stable states in the Introduction (starting lines 60 of the manuscript with track changes).

l126-133 The hypotheses are a bit foggy and based on my opinion not very well introduced in the intro-duction section. The community types studied are not clearly introduced here – it is not clearly said that the effect of restoration works on variously long-lasting encroachment and reference unmanaged grass-lands are compared here. In the first hypotheses is not clear what do you mean with "communities vary in parallel"

We agree with the criticism and have changed the text accordingly. Community types are now clearly introduced (line 148 of the manuscript with track changes). We also rewrote the hypotheses to make them clearer (line 158). We removed the expression 'communities vary in parallel'.

l338-340 This assumption is rather strong. If something changes faster than the other it does not really means that it driver anything – it could be just simply that this type of community/organism group is more sensitive to the changes than the other. Of course in case of encroachment because of direct competition for light you can imply that the shrub encroachment drives the changes in the herb layer, but this is also highly resistant to a particular level of encroachment. This assumption should be down toned a bit.

We completely agree and have changed the text accordingly (starting line 410). We basically address which groups of organisms change more quickly and which more slowly. This makes it possible to put forward hypotheses about which groups might drive others, but it does not indicate causal relationships. 

Reviewer #2: 

This paper compared variations of soil microbial community composition and vegetation composition fol-lowing different stages of woody encroachment and analyzed the possible drivers of these variations. The study may provide valuable information for management of semi-natural calcareous grassland.

Thank you!

Generally, the manuscript was well written. The introduction was logically organized and the discussion part provided sufficient analysis and interpretation of the data. The following are two suggestions for the authors.

Thank you!

（1） The definite conclusions seemed to be missing in both the abstract and in the main text.

We agree and have now attempted to express the conclusions more clearly.

（2） In the Methods part, there were no sufficient information on the vegetation (species identity, abun-dance, canopy height, etc.), climate, and soil etc. for each site sampled.

We appreciate this observation and have added more descriptive information about the study sites.

Thank you once again for providing such a thoughtful and thorough review of our work. We look forward to hearing from you.

Yours sincerely, on behalf of the authors

Ayesh Wipulasena

---

## [Decision Letter · Decision Letter 1]

20 Sep 2023

Soil community composition in dynamic stages of semi-natural calcareous grassland

PONE-D-23-21042R1

Dear Dr. Wipulasena,

We’re pleased to inform you that your manuscript has been judged scientifically suitable for publication and will be formally accepted for publication once it meets all outstanding technical requirements.

Kind regards,

Jian Liu

Academic Editor

PLOS ONE

Additional Editor Comments (optional):

Reviewers' comments:

Reviewer's Responses to Questions

**Comments to the Author**

1. If the authors have adequately addressed your comments raised in a previous round of review and you feel that this manuscript is now acceptable for publication, you may indicate that here to bypass the “Comments to the Author” section, enter your conflict of interest statement in the “Confidential to Editor” section, and submit your "Accept" recommendation.

Reviewer #1: All comments have been addressed

Reviewer #2: All comments have been addressed

2. Is the manuscript technically sound, and do the data support the conclusions?

Reviewer #1: Yes

Reviewer #2: No

3. Has the statistical analysis been performed appropriately and rigorously? 

Reviewer #1: Yes

Reviewer #2: Yes

4. Have the authors made all data underlying the findings in their manuscript fully available?

Reviewer #1: Yes

Reviewer #2: Yes

5. Is the manuscript presented in an intelligible fashion and written in standard English?

Reviewer #1: Yes

Reviewer #2: Yes

6. Review Comments to the Author

Reviewer #1: Thank you for the revision. All of my comments have been addressed. I have no further queries. I suggest accept to the ms.

Reviewer #2: The authors have responded to all my comments and improved the mansucript and the manucript may be suitable for publication now.

7. PLOS authors have the option to publish the peer review history of their article (what does this mean?). If published, this will include your full peer review and any attached files.

Reviewer #1: No

Reviewer #2: No

---

## [Editor Report · Acceptance letter]

9 Oct 2023

PONE-D-23-21042R1 

Soil community composition in dynamic stages of semi-natural calcareous grassland 

Dear Dr. Wipulasena:

I'm pleased to inform you that your manuscript has been deemed suitable for publication in PLOS ONE. Congratulations! Your manuscript is now with our production department. 

Kind regards, 

on behalf of

Dr. Jian Liu 

Academic Editor

PLOS ONE